# The Importance of Iron Status for Young Children in Low- and Middle-Income Countries: A Narrative Review

**DOI:** 10.3390/ph12020059

**Published:** 2019-04-16

**Authors:** Andrew E. Armitage, Diego Moretti

**Affiliations:** 1MRC Human Immunology Unit, MRC Weatherall Institute of Molecular Medicine, University of Oxford, John Radcliffe Hospital, Oxford, OX3 9DS, UK; 2Laboratory of Human Nutrition, Institute of Food Nutrition and Health, Department of Health Sciences and Technology, ETH Zürich, CH-8092 Zürich, Switzerland; diego.moretti@ffhs.ch; 3Nutrition Group, Health Department, Swiss Distance University of Applied Sciences, CH-8105 Regensdorf, Switzerland

**Keywords:** iron, anaemia, infection, malaria, immunity, brain development, growth, microbiome, hepcidin, ferritin, iron supplementation, infants, children, low and middle income countries

## Abstract

Early childhood is characterised by high physiological iron demand to support processes including blood volume expansion, brain development and tissue growth. Iron is also required for other essential functions including the generation of effective immune responses. Adequate iron status is therefore a prerequisite for optimal child development, yet nutritional iron deficiency and inflammation-related iron restriction are widespread amongst young children in low- and middle-income countries (LMICs), meaning iron demands are frequently not met. Consequently, therapeutic iron interventions are commonly recommended. However, iron also influences infection pathogenesis: iron deficiency reduces the risk of malaria, while therapeutic iron may increase susceptibility to malaria, respiratory and gastrointestinal infections, besides reshaping the intestinal microbiome. This means caution should be employed in administering iron interventions to young children in LMIC settings with high infection burdens. In this narrative review, we first examine demand and supply of iron during early childhood, in relation to the molecular understanding of systemic iron control. We then evaluate the importance of iron for distinct aspects of physiology and development, particularly focusing on young LMIC children. We finally discuss the implications and potential for interventions aimed at improving iron status whilst minimising infection-related risks in such settings. Optimal iron intervention strategies will likely need to be individually or setting-specifically adapted according to iron deficiency, inflammation status and infection risk, while maximising iron bioavailability and considering the trade-offs between benefits and risks for different aspects of physiology. The effectiveness of alternative approaches not centred around nutritional iron interventions for children should also be thoroughly evaluated: these include direct targeting of common causes of infection/inflammation, and maternal iron administration during pregnancy.

## 1. Introduction

Young children growing up in low- and middle-income countries (LMICs) are frequently exposed to concurrent physiological challenges and environmental hazards that may influence both their current health and future development. In such settings, a high infection risk often co-exists with nutritional deficiencies, amongst which iron deficiency (ID) is of major importance. ID is best known for causing anaemia, affecting hundreds of millions of young children, particularly in LMICs [1]. Iron deficiency anaemia (IDA) affects over 1.2 billion people globally and was recently classified as the leading cause of years lived with disability (YLDs) in low/low–middle socioeconomic settings [2]. However, ID also occurs frequently without manifesting anaemia, meaning the number of people affected globally will be considerably higher. Importantly, the clinical and developmental impacts of iron deficiency also go beyond anaemia [3].

In this narrative review, we discuss the biology of iron in early childhood, with a particular focus on children living in settings with high burdens of infection, nutritional deficiency and anaemia. We aim to identify research gaps, controversies and accepted principles from biological, nutritional and epidemiological points of view. In the first section, we consider why iron is required during the early years of life, how iron status is defined, and how iron handling is regulated. In the second section, we examine the implications of iron status and therapeutic iron interventions for different physiological and developmental processes of particular relevance in early childhood. The final section summarises factors that should be accounted for when planning interventions aimed at adjusting iron status in such populations.

## 2. Regulation of Iron Status during Early Childhood

### 2.1. Iron Demand in Early Childhood

Iron is widely used as a cofactor in cellular biochemistry owing to its ability to act as an electron donor or acceptor. The numerous examples of proteins that use haem prosthetic groups or iron-sulphur clusters as cofactors, or which directly coordinate iron for catalytic functions, demonstrate the wide utility of iron in biology. Box 1 provides a far from exhaustive list of examples of how such proteins are used across key cellular processes. An adequate supply of iron is thus required to support normal physiology.

The daily requirement for iron is greater in early childhood than during adulthood [4,5]. Adulthood requires a “steady-state” turn-over of iron to maintain tissue homeostasis while ensuring flexibility to adapt iron handling according to changes in physiological or environmental circumstances (e.g., blood loss, infection, pregnancy, nutritional limitations, changes in altitude) [6]. In contrast to adults, young children must expand their blood volume, increase muscle and tissue mass, and are at a key stage in terms of neurodevelopment and brain growth: growing tissues therefore require an absolute increase in iron content to maintain a state of iron repletion.

Box 1Examples of iron involvement in cellular biochemistry.
The haem group of oxygen-transporting haemoglobins is the major and best-known destination of absorbed iron, supporting adequate erythropoiesis, with an estimated two-thirds of body iron found within erythrocyte haemoglobin.Haem moieties, in which iron is coordinated within protoporphyrin IX, also mediate oxygen-storage by myoglobin in muscle, and are used by cytochrome P450 monooxygenases, and by cytochrome c and cytochrome c oxidase in the mitochondrial electron transport chain for oxidative ATP production. The final step of haem biosynthesis is itself iron dependent, catalysed by the Fe_2_-S_2_-containing protein ferrochelatase. Iron-sulphur clusters are central as cofactors in energy production, being complexed within mitochondrial aconitase in the tricarboxylic acid (TCA) cycle, and complexes I–III in the electron transport chain. They are used by DNA polymerases, DNA primase subunits, and DNA helicases reflecting involvement in DNA replication and repair. Nucleotide biosynthesis provides the substrate for DNA replication and depends on ribonucleotide reductase (RnR), a di-iron monooxygenase. Iron(II)- and 2-oxogluatarate-dependent dioxygenases include the prolyl hydroxylases PHD-1, 2 and 3 which regulate the Hypoxia-Inducible Factor (HIF)-dependent response to hypoxia, and the TET family of methylcytosine dioxygenases which are involved in histone demethylation and consequent epigenetic regulation.Iron is centrally involved in the antimicrobial oxidative burst employed during neutrophil responses to infection, involving the haem-dependent NADPH oxidase, and iron-dependent myeloperoxidase.


### 2.2. Iron Supply in Early Childhood

#### 2.2.1. The Maternal Iron Endowment

Neonates are born with an endowment of iron from their mother, estimated as approximately 75 mg/kg (approximately 260 mg total body iron) in a healthy full-term neonate [7]. The size of the iron endowment is strongly influenced by gestational age, birthweight and timing of cord clamping [7]. Maternal iron status does not appear to predict neonatal iron status during maternal iron sufficiency [8,9] or in the presence of maternal iron-restricted erythropoiesis [10]. However, several studies report compromised infant iron status following maternal iron deficiency anaemia (e.g., [11,12]). This suggests that unless maternal iron is severely limited, foetuses tend to acquire maternal iron efficiently during gestation irrespective of maternal iron status. The size of the maternal iron endowment and its consumption rate vary between infants, with declines in iron status greater in boys than girls [13,14]. A significant proportion of maternal–foetal iron transfer occurs during the final weeks of gestation [9]. Consequently, pre-term infants are at high risk of earlier onset ID. Similarly, since most of the iron is contained within haemoglobin, lower birthweight infants—who have a smaller total blood volume—carry lower absolute body iron content and are at higher risk of ID [13]. Delaying cord clamping, which can enable blood volume increases of ~40 mL/kg birthweight, is associated with significantly improved iron status and lower ID risk during the first year (reviewed in [7]).

The maternal iron endowment is thought to sustain the iron requirements of an exclusively breastfed healthy term infant for up to half of the first year [7]. Since iron is utilised to support growth and development, the initial iron stores become depleted as infancy progresses, with steeper declines in more rapidly growing infants [13,15,16,17]. This means iron must be obtained from exogenous sources including complementary foods and via food fortification or supplementation [7]. Human breastmilk is often considered to contain highly bioavailable iron but only in small quantities that are insufficient to sustain iron requirements as infancy progresses [18,19,20]. Therefore, weaning diets must contain adequate sources of bioavailable iron.

#### 2.2.2. Dietary Iron: Differences between High- and Low-Income Settings

Two categories of dietary iron exist: haem iron, present in animal-source foods, is highly bioavailable (~20–50% of ingested iron is absorbed); in contrast, non-haem iron, which accounts for a higher proportion of dietary intake, has highly variable but typically substantially lower bioavailability [21]. Absorption of non-haem iron, unlike haem iron, is influenced by the presence of dietary enhancers (e.g., ascorbic acid (AA) and muscle protein) and inhibitors (e.g., phytates and polyphenols) (Reviewed in [22]). Based on overall dietary composition, iron bioavailability can be classified as poor (average fractional absorption (FA) <5%), intermediate (FA, 5–15%) or high (FA, >15%). Bioavailability relates to the relative balance of unrefined cereals and pulses (high in inhibitory phytates), fresh fruits and vegetables (sources of AA), refined cereals (lower in phytates), and meat, fish and poultry (rich in haem iron and muscle protein). Weaning foods should possess high iron bioavailability, as the iron/energy requirement ratio is particularly elevated in infancy. These factors likely contribute to differences in iron status between young children from higher-income settings, where fortification of weaning diets, cereals and meat-based diets are commonplace, compared with those from LMICs where diets often contain lower absolute iron content, combined with lower dietary bioavailability enhancer:inhibitor ratios.

The recommended nutrient intake (RNI) of iron for 7–12-month-old infants (i.e., required quantity of absorbed iron) has been estimated as 0.69 mg/day (97.5th percentile, 1.07 mg/day) [5,23]. If bioavailability of dietary iron is below 10%, intake of >10.7 mg/day would be required to guarantee reaching the 97.5th RNI percentile, which may present a challenge in resource-limited settings; accordingly, children may frequently fail to meet the RNI, and ID may ensue [23]. Risk of nutritional ID can be classified via markers of dietary exposure relating to both iron intake and overall iron dietary bioavailability (reviewed in detail by the Biomarkers of Nutrition for Development (BOND) project [24]). However, dietary intake and bioavailability assessment remains challenging, with reliable intake estimates requiring 7–10 days of weighed food records. Nonetheless, at the individual level, dietary counselling and advice to increase iron intakes [25] and bioavailability may concomitantly increase the likelihood of RNI being met [26]. Programmatic evidence for a benefit of such a strategy is lacking yet would be highly desirable.

Differences between high- and low-income settings in intake of bioavailable iron could become further accentuated over the coming decades in relation to anthropogenic increases in atmospheric CO_2_ concentrations. The absolute iron content of widely consumed crops, specifically C3 grasses (wheat and rice) and legumes (field peas and soybeans), decreases with increasing CO_2_ [27], and modelling suggests this would translate to an increase in global iron deficiency prevalence, focussed particularly on LMIC populations for whom these cereals constitute dietary staples [28]. Climate change is also predicted to contribute significantly to a declining fish catch over the coming decades, reducing consumption of highly bioavailable iron and increasing micronutrient deficiency in coastal communities, again predominantly in LMIC settings [29].

As discussed below, iron uptake may also be inhibited by infection and inflammation for which the burden is higher in LMICs compared to higher-income countries. Furthermore, young children from LMIC populations are at higher risk of parasitic (most notably hookworm) infection, which is associated with intestinal blood loss, and therefore additional iron depletion [30,31]. Together, these factors illustrate the greater challenge faced by young children in LMIC settings in maintaining adequate iron status to sustain growth and development, in comparison to children of similar age in higher income settings.

### 2.3. Molecular Control of Iron Handling

In understanding the relationships between iron deficiency, iron absorption and inflammation, and in seeking to develop safe and effective interventional strategies, it is important to appreciate the molecular basis of iron control. While the redox properties of iron make it useful for cellular biochemistry, unchaperoned iron can catalyse the generation of toxic free radicals. Iron transport and storage must therefore be carefully regulated to ensure it is effectively targeted and employed in a controlled manner. Details of cellular and systemic iron handling have been reviewed in detail recently [6]. It has been long-established that systemic iron homeostasis is maintained through regulation of absorption, with iron losses not specifically regulated [32]. Iron sensing mechanisms must ensure non-regulated losses (e.g., through bleeding, intestinal cellular sloughing, or helminth infection) are replaced through appropriate absorption. When control of iron absorption is lost, either iron loading or iron deficiency may ensue: failures to prevent excess iron absorption, as in hereditary haemochromatosis or during non-transfusion-dependent beta-thalassaemia, lead to gradual toxic iron accumulation in the liver and other organs. Conversely, failure to absorb sufficient amounts of bioavailable iron eventually leads to anaemia, as in dietary iron deficiency, during chronic inflammation, and in the rare heritable iron-refractory iron deficiency anaemia (IRIDA) [6].

Iron balance must therefore be carefully maintained. This is achieved by coordinating cellular and systemic iron homeostasis [33]. Posttranscriptional regulation by the IRE/IRP (iron response element/iron regulatory protein) system is central in the control of cellular iron homeostasis. IRPs are RNA binding proteins which, during cellular iron deficiency, either stabilise iron-related mRNAs (e.g., *TFR1*, for iron acquisition) by binding 3’UTR IREs, or mediate translational repression of mRNAs by binding 5’UTRs (e.g., *FPN1*, for iron export; *FTH1*, *FTL* encoding ferritin heavy and light chains respectively for iron storage) [6]. Systemic iron homeostasis on the other hand centres on the regulation and activity of the iron regulatory hormone hepcidin [34].

#### 2.3.1. The Hepcidin–Ferroportin Interaction

Hepcidin coordinates both iron uptake through duodenal enterocytes and iron recycling (primarily derived from senescent erythrocytes) by reticuloendothelial macrophages. Hepcidin binds the iron exporter ferroportin, which is highly expressed on these cells, triggering its internalisation and ubiquitin-mediated degradation; this prevents the export of cellular iron into circulation [35]. Hepcidin-binding also directly occludes the ferroportin ion channel, providing an additional inhibitory mechanism [36]. Thus, when hepcidin concentrations are high, dietary iron uptake is inhibited, iron becomes sequestered in ferroportin-expressing macrophages, and serum iron concentrations decline. Hepcidin acts very rapidly: serum iron falls by ~80% within an hour of hepcidin injection into mice [37]. The erythroid compartment also contributes to systemic iron homeostasis by returning iron to circulation: hepcidin-sensitive ferroportin is expressed by erythroblasts and is found on mature erythrocyte membranes [38,39]; erythroid-specific ferroportin knockout mice display mild serum iron deficiency, consistent with a proportion of circulating iron being derived from the erythron [40].

Inherited disorders of iron homeostasis reveal the centrality of the hepcidin–ferroportin interaction in maintaining iron balance. Iron overloading in haemochromatosis can be caused by mutation in the gene encoding hepcidin (*HAMP*) itself. More frequently, it links to mutations in genes encoding components of the signalling pathways mediating hepcidin upregulation in response to iron (including *HFE, HJV*, *TfR2*, *BMP6*) [41,42]. Hepcidin insufficiency characterises each of these conditions. Likewise, patients with “gain-of-function” ferroportin mutations (rendering ferroportin hepcidin-resistant) develop iron overload [43,44]. Patients with “loss-of-function” ferroportin disease display a distinct disordered iron phenotype (including macrophage rather than parenchymal iron loading) [45]. Contrastingly, IRIDA results from loss-of-function mutations in *TMPRSS6*, which encodes the protein matriptase-2 that normally inhibits the production of hepcidin [46].

#### 2.3.2. Hepcidin Regulation by Iron

Details of the molecular pathways regulating hepcidin have been reviewed extensively elsewhere [34,47]. Broadly speaking, to maintain homeostasis, hepcidin is upregulated in hepatocytes in response to sensing increased iron via the BMP–SMAD (Bone Morphogenetic Protein—Sons of Mothers against Decapentaplegic) pathway [47]. Liver Sinusoidal Endothelial Cell (LSEC)-derived BMP6 and BMP2 are key in this process [48,49,50,51]. BMP6 is more responsive than BMP2 to tissue iron loading [52], although how iron sensing leads to BMP6 upregulation is yet to be described. Hemojuvelin (HJV) is a key accessory receptor that enhances BMP-mediated signalling [53]. Sensing of acute changes in serum iron concentration and transferrin saturation also converges on the BMP–SMAD pathway, involving transferrin receptor 1 (TfR1/CD71), transferrin receptor 2 (TfR2), the haemochromatosis protein HFE and BMP receptors (although not requiring BMP6 induction) (reviewed in [47]).

#### 2.3.3. Hepcidin Regulation during Inflammation

Hepcidin is also induced during the acute phase response to inflammatory stimuli [54,55]. Upregulation is primarily mediated via the JAK–STAT3 (Janus Kinase—Signal Transducer and Activator of Transcription proteins 3) pathway, most notably by interleukin-6 (IL-6) [56,57,58], but probably also by other cytokines including IL-22, type I interferon, and IL-1beta [55,59,60,61]. Cross-talk with BMP6/SMAD signalling is also required for the normal response to inflammatory stimuli, since loss of HJV blunts the hepcidin response in this context [62]. Activin B, which also signals via SMAD pathways, is induced during inflammation/infection and can upregulate hepcidin in vitro, although its role in inflammatory hepcidin induction in vivo remains uncertain [63,64,65,66,67]. Hepcidin activity during infection and inflammation leads to characteristic hypoferremia [68], shown to be protective against extracellular siderophilic bacterial infections [69,70]; hepcidin-independent mechanisms including the transcriptional downregulation of ferroportin expression also likely contribute [71]. Prolonged inflammatory stimulation of hepcidin and consequent hypoferremia contributes to the pathogenesis of anaemia of inflammation (AI) by limiting iron supply to the erythron [72].

#### 2.3.4. Hepcidin Suppression during Erythropoietic Demand

Erythropoiesis is highly iron-demanding, especially when expanded physiologically in response to acute blood loss or when it is ineffective as during beta-thalassaemia. Erythropoietic expansion is stimulated by erythropoietin signalling, whose activity induces the production of the hepcidin suppressive hormone erythroferrone in erythroblasts [73]. Erythroferrone suppresses liver hepcidin production through binding and inhibiting BMP6 and the related BMP family members, BMP5 and BMP7 [74] (which themselves are upregulated by iron in the absence of BMP6 [75]). Iron consumption by enhanced erythropoiesis also reduces the hepcidin-stimulatory signal from serum iron [76,77]. Similarly, during hypoxia, hepcidin can be suppressed through the cAMP Response Element Binding protein (CREB/CREB-H) pathways in response to high concentrations of Platelet-Derived Growth Factor-BB (PDGF-BB) activity [78]. Together, these mechanisms inhibit hepcidin production, liberating iron from hepatic and macrophage stores and facilitating iron absorption to supply erythropoiesis. Moreover, duodenal iron absorption is further promoted when hepcidin is low since ferroportin-dependent iron flux through enterocytes leads to stabilisation of the hypoxia-inducible factor HIF-2alpha leading to the upregulation of genes involved in iron absorption (*Dcytb*, *Dmt1*, and *Slc40a1*—encoding ferroportin itself) [79].

In summary, multiple signals derived from diverse physiological systems are integrated at the level of hepcidin regulation to ensure adequate iron supply whilst mitigating against the dangers of iron loading and, as discussed below, acute siderophilic infection.

### 2.4. Regulation of Hepcidin and Iron Status in Infancy and Early Childhood

Concentrations of hepcidin, although variable, are relatively high in the first few months of life, when iron availability is not limited; this is consistent with the regulation of hepcidin operating in infancy in response to iron status as in adults [17,80]. Evidence has been presented however that in the first 6 months, absorption of iron supplements does not correlate with iron status, suggesting dietary iron absorption may bypass hepcidin-mediated regulation at this point [18,19,81]. Nonetheless, most of the iron entering circulation is derived from iron-recycling erythrophagocytic macrophages. An acute hypoferremia in neonates occurs during the first 48 hours of life [82,83], which is associated with raised hepcidin (Prentice, A.M.; et al., personal communication). This is consistent with hepcidin-mediated control of macrophage iron release being functional even in neonates, despite potential differences in intestinal iron absorption between early and later stages of infancy.

Studies in Gambian infants reveal profound reduction in serum iron concentration during the first months of life [17], to levels well below those typical in infants from high-income settings [84]. Ferritin decreases through the first year, more profoundly in males, as stores are consumed for expanding the erythron and growth and development of tissues (as discussed below) [8,14]. Related decreases in hepcidin also occur in African infants, again more notably in boys than girls [80,85,86], with the extent of weight gain dominantly predicting the extent of hepcidin and ferritin decline [17]. Hepcidin concentrations in African pre-school children are associated with other biomarkers of iron status, inflammation and erythropoietic demand, suggesting hepcidin is regulated similarly in young children and adults [87].

In contrast to the studies in African infants, declines in hepcidin during the first year were not observed in two studies of European infants [88,89]. This may relate to higher contents of bioavailable iron in typical European weaning diets. A recent study in Gambian infants (6–27 months) found that respiratory infections and fever incidence (but not diarrhoea or faecal calprotectin—a marker of intestinal inflammation) were strongly associated with raised hepcidin; inflammation (marked by CRP), even if very low-grade, was the dominant predictor of hepcidin [90]. Chronic, even mild, inflammation may therefore play a key role in inhibiting dietary iron uptake, contributing to the declining iron status of early childhood in such settings.

### 2.5. Classification of Iron Deficiency

Before discussing how low iron status influences key physiological processes in young LMIC children, it is useful to review how iron status can be classified. Haemoglobin (Hb) is a standardized and accessible marker whose concentration is directly proportional to oxygen carrying capacity. However, it is a poor indicator of ID with low sensitivity (ability to identify iron deficiency when present) and specificity (ability to correctly identify subjects without deficiency). This reflects anaemia being a late manifestation of ID and having multiple aetiologies, especially in LMIC children. Thus, iron status must be assessed using more sensitive and specific biomarkers. Ideally, these should identify different degrees of iron status, ranging from depletion of iron stores, to iron-deficient erythropoiesis without anaemia, to iron-deficiency anaemia [24].

The gold standard approaches for assessing ID are bone marrow iron (Prussian blue) staining and, in the case of anaemia, evaluation of the haematological response to therapeutic iron [91]. Bone marrow staining reveals absence of reticuloendothelial iron storage and therefore true ID. Neither of these are appropriate for routine clinical or epidemiological assessment and, without an individually effective biochemical biomarker of iron balance, classification of ID is challenging, especially in LMIC children.

#### 2.5.1. Ferritin and Iron Stores

Serum ferritin is linearly predictive of iron stores in the absence of inflammation, with 1 µg/L corresponding to 120 µg storage iron/kg body weight [92]. It reflects intracellular iron content of macrophages and hepatocytes. The commonly used cut-off of 12–15 µg/L is diagnostic for iron deficiency, but it has low sensitivity. Consequently, appropriate cut-offs for use in different settings and purposes are commonly debated; for example, higher cut-offs may be advisable for confirming a diagnosis (avoiding false negatives) compared to those used for population screening (where it is desirable to avoid false positives) [91]. Moreover, ferritin is an acute phase protein, induced during inflammation, infection and liver damage. Promising approaches that adjust ferritin values according to concurrent inflammation have recently been proposed [93], but will require further validation. The characteristics of iron metabolism in infancy and early childhood may return different thresholds for defining deficiency especially in settings with high infectious disease pressure, and clearly more evidence is required [94,95].

#### 2.5.2. Markers of Iron-Restricted Erythropoiesis

The soluble transferrin receptor (sTfR1) represents a detectable circulating truncated form of the cellular iron uptake receptor, TfR1, increasingly expressed once tissue iron supply becomes scarce. Other than during the first few months of life, the bone marrow is the most avid iron consumer, and sTfR1 concentration is generally thought to reflect iron demand from erythroid precursor cells. It is less affected than ferritin by inflammation and infection, but may be modified by conditions affecting erythropoiesis, including thalassaemia and sickle cell anaemia [96]. The ratio between sTfR and log-ferritin (ferritin index) offers the advantage of combining two markers covering different aspects in iron depletion, generally improving sensitivity and specificity compared to their individual use [24]. Of note, paper-based smartphone-app-guided point-of-care assays for ferritin and sTfR show promise in proof-of-principle studies, suggesting field measurement of these markers in children from resource-limited settings may become feasible [97,98]. Other markers of iron-restricted erythropoiesis are zinc protoporphyrin (ZPP) and the ZPP/haem ratio in red blood cells; ZPP accumulates in the last step of haemoglobin synthesis when iron is lacking [99]. Certain red cell indices may also be useful as markers of iron deficient erythropoiesis [24], among which reticulocyte haemoglobin content (Ret-Hb/CHr) is prominent [100,101] and included in the American Academy of Pediatrics guidelines for the evaluation of anaemia [24]. Red cell indices are reported to be more suitable than ferritin alone in predicting response to intravenous iron in patients with chronic kidney disease [102], but their performance in detecting nutritional anaemia in childhood has not been evaluated systematically [24].

#### 2.5.3. Serum Iron and Transferrin Saturation

Serum iron and transferrin saturation (Tsat, the proportion of iron-binding sites of the iron-transport protein transferrin occupied by iron) reflect iron availability for tissues at any given moment, and thus potentially provide information on peripheral iron delivery. However, they are affected by circadian rhythm and decreased by inflammation and infection, rendering their interpretation challenging in young children [24]. Nevertheless, extremely low serum iron concentrations described in concert with low ferritin/hepcidin in African infants, well below the typical range in infants elsewhere, does likely reflect a severe deficiency in iron supply for iron-requiring tissues [17,84].

#### 2.5.4. Hepcidin

Hepcidin shows promise as a further reliable marker to identify ID (during which it is suppressed), and to distinguish IDA from inflammatory anaemia in settings with high infection pressure [103]. It had moderate performance in predicting bone marrow ID in severely anaemic children in Malawi when compared to ferritin index [104]. As the hormone that regulates iron uptake, it also predicts oral iron responsiveness [103,105,106]. Like ferritin, hepcidin is an acute phase protein potentially confounded by inflammation [54,55] but, unlike ferritin, hepcidin synthesis is directly suppressed by expanded erythropoiesis. Hepcidin also shows diurnal variation [107]. High quality hepcidin assays are increasingly available and accessible, with efforts being made towards their harmonisation and standardisation [108,109]. With increased understanding of hepcidin biology and further analytical refinements, hepcidin may become more frequently used in assessing iron status [110], including in young children.

The extent to which these serum biomarkers indicate true iron deficiency must be considered. A recent study assessed the performance of biochemical iron status markers in predicting bone marrow ID in 1–5-year-old anaemic children in Mozambique, a setting with a high burden of infection/inflammation. As expected, serum ferritin displayed poor sensitivity but 100% specificity (zero false positives) in identifying bone marrow ID. In contrast, sTfR1 showed 83% sensitivity and 50% specificity. Combining the two markers into the ratio, resulted in improved performance, but overall, even the best-performing markers failed to identify ~25% of children with ID correctly [111]. The standardization of assays across different suppliers and instruments is an issue for all iron status markers, in particular for sTfR1 and hepcidin. Further research to improve interpretation, accessibility and analytics of current and experimental markers of iron status in settings with high infection pressure is clearly a research priority.

## 3. The Importance of Iron Status for Different Physiological Systems in Early Childhood

### 3.1. Iron and Oxygen Delivery

#### 3.1.1. Erythropoiesis in Early Childhood

Anaemia refers to a state in which the oxygen-carrying capacity of blood, mediated by haemoglobin, falls below the level required to support normal physiological functions. In a healthy neonate, approximately 70% of total body iron (estimated as 234–334 mg, depending on birthweight and cord clamping time) is contained within haemoglobin, which is more concentrated in neonates than in adults (typically ~170g/L, calculated to contain 155 mg iron in a 3.5 kg neonate) [4]. Higher haemoglobin concentrations at birth reflect a relatively hypoxic environment in utero. The transition to a normoxic environment after birth leads to a marked decline in erythropoietic output. Combined with shorter erythrocyte lifespan and dilution effects related to growth, haemoglobin concentrations fall notably in the first 1–2 months [112]. This “physiological anaemia of infancy” is accompanied by redistribution of iron from senescent red cells to iron stores (indicated by ferritin concentrations) [113]. From 6–8 weeks, erythropoietin concentrations and erythropoietic output increase, typically raising haemoglobin concentrations modestly [112]. The daily iron demand for normal erythropoietic output requires considerably higher quantities than are absorbed each day, bearing in mind the low iron content of breastmilk. Thus, most iron is supplied from iron stores derived from recycling of senescent red cells by the reticuloendothelial system.

As infancy progresses into early childhood, failure to meet the iron demand of growth-related blood volume expansion can lead to iron-restricted erythropoiesis. This is commonly observed in young children from LMICs. Failure in iron supply can result from either nutritional iron deficiency, or physiological restriction of macrophage iron recycling and iron absorption (functional iron deficiency), or commonly both. Iron restriction is a feature of the response to infection and inflammation, relating to elevated hepcidin concentrations and suppressed ferroportin activity. Erythropoietic suppression and increased erythrocyte turnover rate also likely contribute to anaemia pathogenesis during inflammation [72].

#### 3.1.2. Molecular Interplay between Iron Handling and Erythropoiesis

Erythropoiesis is sensitive to decreased iron availability [114]. With decreasing transferrin saturation, ex vivo erythroblast differentiation is diminished relative to granulopoiesis and megakaryopoiesis, involving aconitase-mediated regulation [115]. Transferrin receptor 2 (TfR2) contributes to hepcidin induction in response to circulating iron in hepatocytes but is also expressed on erythroblasts. Erythroblast TfR2 interacts with the erythropoietin receptor (EpoR) to modulate sensitivity to Epo signalling via interactions with Scribble, a key regulator of cellular trafficking and signalling [116,117,118,119]. Together, this provides a plausible mechanism by which erythropoietic output can be adjusted during iron deficiency (reviewed in [120]). Moreover, as stated above, erythroblasts express hepcidin-sensitive ferroportin (from a transcript (*FPN1B*) that is not subject to IRP-mediated regulation) [38,39]. Erythroid-specific ferroportin knockout mice displayed mild serum iron deficiency consistent with a contribution of erythroid iron to the circulating iron pool [40]. During iron deficiency (when hepcidin concentrations are low), this has been proposed as a safety mechanism to ensure supply of iron to iron-sensitive tissues at the expense of erythropoiesis, also protecting erythrocytes from oxidative stress [40].

#### 3.1.3. The Burden of Anaemia in LMIC Children

Anaemia is typically defined by haemoglobin concentrations falling below specific thresholds. The World Health Organisation (WHO) recommends thresholds of 100–109 g/L, 70–99 g/L and <70 g/L to identify mild, moderate and severe anaemia respectively in children under 5 years of age [121]. However, it should be noted that a re-evaluation of the use and interpretation of anaemia-defining haemoglobin thresholds is underway [122]. Anaemia was estimated to affect 273 million children globally in 2011, disproportionately exerting its burden in low-income settings [123]. Iron deficiency is often purported to account for about half of the anaemia burden, yet estimates of the proportion of anaemia that is iron responsive are lower than this (42% globally, and only 32% in Africa) [1,124]. These estimates point to the diverse aetiologies of anaemia: other causes include inflammation and infection (notably malaria), inherited erythrocyte disorders (e.g., sickle cell anaemia, thalassaemia, Glucose-6-phosphate dehydrogenase (G6PD) deficiency), and other micronutrient deficiencies (e.g., Vitamin A, Vitamin B12).

Chronic but mild iron-deficiency anaemia may not manifest symptomatically, and could even provide benefit in terms of reduced malarial risk (discussed further below) [125]. However, while anaemia in young children could lead to lethargy, it may more importantly influence longer-term developmental outcomes [126].

### 3.2. Iron and Neurological Development

Iron deficiency, both with and without anaemia, is associated with impaired cognitive, behavioural and motor function development [3,127,128]. Consequently, low iron status may have long-term effects, for example affecting educational attainment and career potential later in life [129]. Iron requirements in the developing brain are considered spatially and temporally sensitive [130,131]. A “scaffolding” process during development has been suggested, where each developmental stage depends on completion of the previous one [132]. This underlies the concern for ensuring adequate iron status during the critical first 3 years of life [126].

As discussed above, erythropoiesis is sensitive to iron status. Based on animal models, it has been suggested that erythroid iron is rechannelled back to circulation during iron limitation for prioritised use by other tissues, including brain, [39,40]. However, other studies suggest that iron limitation may lead to brain iron deficiency prior to the development of frank anaemia, suggesting erythropoiesis may not be compromised first [100]. After iron deprivation, decreased brain iron that preceded anaemia was reported in phlebotomized lambs [133], in human infants born from diabetic mothers [134], and in sheep affected by intrauterine hypoxemia [135]. During repletion, decreased brain iron levels persisted in infant rats [136] and haemoglobin repletion preceded brain iron accrual [137].

#### 3.2.1. Roles for Iron in Brain Development

Normal neurological development involves an array of iron- and haem-dependent proteins. Studies, primarily using rodent models, demonstrate links between ID and multiple neurodevelopmental impairments. These range from decreased axon myelination and impaired monoamine and gamma-aminobutyric acid (GABA) neurotransmitter system development to reduced hippocampal function and compromised central nervous system (CNS) energy metabolism (reviewed in [3,100,138]). The specific brain region or process affected is likely to depend on the timing of ID, with those undergoing most rapid growth and development concurrent with ID principally affected [138]. IDA could also indirectly affect the CNS by impairing thyroid function [139] or by increasing the risk of lead poisoning [140,141], both of which are associated with compromised neurological development.

Tools for investigating the impact of brain region-specific iron deficiency in the absence of anaemia are emerging. For example, a mouse model with reversible (Tet-OFF) hippocampal TfR1 deficiency enables comparison of chronic hippocampal ID with reversed early-life hippocampal ID [142]; restoration of hippocampal TfR1 expression at postnatal day 21 (heuristically corresponding to ~2 years of age in human infants) reversed some, but not all, of the gene expression changes associated with ID [142]. This further points to lasting impact of early-life ID on neurodevelopmental phenotype, and calls for further investigation into optimal timings of interventions aimed at restoring iron status in infancy [143].

#### 3.2.2. Iron Interventions and Cognitive Outcomes

In population studies, meta-analyses of the effect of iron supplementation on cognition in young children have yielded ambiguous results. No significant collective detectable benefit was reported in one recent meta-analysis (mean difference 1.65 [95% Confidence Interval (CI): −0.63, 3.94], *p* = 0.16; 6 trials; children aged 4–23 months) [144]. This could reflect a genuine lack of effect, or alternatively the inability of the study designs to adequately detect a true effect. Assessing cognitive performance is challenging in young children: test methods typically designed to detect “macroscopic” neurological deficits during early perinatal life may lack sensitivity for subtle changes in performance [145]. In contrast, several studies indicate the benefits of iron supplementation on cognitive function in older children [145,146,147], in particular following longer-term supplementation. Within this discussion, it should also be borne in mind that iron-replete Chilean infants who received high iron formulas displayed reduced cognitive performance at 10 years of age [148].

Together, these analyses highlight the general physiological importance of correcting ID for cognitive development, while at the same time calling for further detailed investigation during early perinatal life. Indeed, few high quality adequately-powered placebo-controlled trials in low-income countries aimed at assessing the impact of iron interventions on cognitive function have been performed [146]. Such data would be highly desirable for informing global health policies regarding universal iron intervention in young children; one such trial—BRISC (“Benefits and Risks of Iron interventionS in Children”)—is currently ongoing in Bangladesh [149].

### 3.3. Iron and Growth

While more rapid growth is associated with greater decline in iron status [13,15,16,17], conflicting evidence exists on the effect of iron status and supplementation on growth. Severe iron deficiency impairs thyroid function [139], and as discussed below, iron status may influence the likelihood of acquiring growth-limiting infections. A limited number of studies, in particular in IDA populations, have found positive effects of iron on growth [150,151] including a recent quasi-randomized study in Ethiopia [152]. However, most studies find no effect [153], with several reporting detrimental growth outcomes in iron-replete infants and children [151,154,155]. A recent meta-analysis reported no overall effect of iron supplementation on length and length-for-age, but a detrimental effect on length gain [144]. The physiological mechanisms underlying these observations require further research. Iron supplementation could inhibit zinc absorption, resulting in reduced growth [19]. Unnecessary supplementation in iron-replete children could also promote oxidative stress via non-transferrin bound iron (NTBI) generation [156], or gut inflammation and dysbiosis (discussed further below) [157,158], again possibly affecting growth. In summary, while growth rate clearly impacts upon iron status, iron status and interventions are not reciprocally or straightforwardly associated with subsequent growth. In particular, the possible relationship of iron interventions with reduced growth in initially iron-replete infants should be investigated further and considered as a component of planning intervention programmes [159].

### 3.4. Iron and Immunity

Vaccination programmes significantly contribute to reducing childhood mortality. However, 5.8 million (95% CI: 5.7, 6.0) under-5 year olds died in 2015, mostly in LMICs, with infections as the major cause [160]. Despite the success of immunization, determinants of both immune system development and efficacious immune responses in infants, especially those from LMICs, are not fully understood. Immune responses to vaccines and natural antigen challenges may differ between lower and higher income countries [161,162]. Understanding childhood immunity specifically in LMICs, in comparison with more developed settings, is therefore of great importance. Inadequate nutritional status, including iron deficiency, represents a plausible contributory factor to impaired immune function in such contexts. Probing immune function in young LMIC children is challenging [163], yet for these populations that are most susceptible to infection-related mortality, even modest enhancement of immunity achieved through novel approaches could bring important benefits [164].

#### 3.4.1. Iron and Immune Ontogeny

The early-life immune system is not static, or equivalent to the adult immune system, but undergoes dynamic age-related changes during ontogeny [163]. This was recently shown with great resolution in Swedish neonates using multidimensional mass cytometry analyses [165]. Non-heritable environmental factors (including age, cytomegalovirus (CMV) status, cohabitation) account for a large part of the inter-individual variation in adult immune phenotypic profiles [166,167]; environmental factors including nutritional status very likely influence early childhood immune development [164]. Iron status and iron supplementation may therefore constitute environmental determinants of immune variation in infants, potentially influencing phenotypic profiles later in life. Effects of iron could be direct (e.g., through affecting lymphopoiesis, which depends on TfR1 suggesting iron requirement [168]) or indirect through modifying the intestinal microbiota [169], which can influence systemic immune phenotypes [170].

#### 3.4.2. Iron and Innate Immune Responses

Altered cellular iron status impacts upon several aspects of the innate immune response [171]. Key effector functions of innate cells such as neutrophils are iron dependent; for example, Saha, et al. recently showed that the bacterial iron-binding siderophore Enterochelin impaired neutrophil antibacterial functions (neutrophil extracellular trap and reactive oxygen species (ROS) generation) by chelating iron (see Box 1) [172]. Cellular iron status also influences macrophage polarization, response to interferon-gamma, and microbicidal effector functions, reviewed in detail elsewhere (e.g., [173,174]). Less precisely characterized is the influence of iron on adaptive immune responses.

#### 3.4.3. Iron and the Adaptive Immune Response

Effective adaptive immune responses involve rapid proliferative expansion of antigen-specific effector T and B cells, accompanied by characteristic metabolic changes that are influenced by nutrient availability [175]. The role of nutrients such as amino acids in supporting normal lymphocyte responses is well-appreciated (e.g., [176]). Key aspects of cellular proliferation and metabolic activity are iron dependent. However, definitive conclusions regarding the precise role of iron in antigen-specific adaptive immunity have remained scarce despite many reports over the last 50 years [177]. Insight from experimental studies has been limited to crude responses to polyclonal stimuli or bulk IgG production, rather than antigen-specific responses, and most studies lack the depth of insight now attainable from more up-to-date immunological methodology. Furthermore, many clinical studies of iron involvement in immunity are inadequately powered, poorly controlled, or potentially confounded by co-morbidities of iron deficiency in resource-limited settings [177].

Iron restriction restricts the proliferation of lymphocytic cell lines in vitro and induces expression of TfR1 (CD71), consistent with increased cellular iron demand [178]. Correspondingly, antibody-mediated blocking of TfR1 function inhibits lymphocyte proliferation [179]. Lymphocytes from iron-deficient elderly individuals showed less proliferative capacity than those from iron-replete controls [180]. In other studies, ID has also been shown to influence T cell numbers, the response to mitogenic stimulation and IL-2 production [181,182]. Conversely, iron supplementation has been associated with restoration of T cell numbers in children [183]. In mice, ID limited T cell-mediated inflammatory liver injury following Concanavilin-A administration [184]. Moreover, a recent study by Wang, et al. [185] demonstrated that iron promoted proinflammatory cytokine expression in T cells by preventing the degradation of the RNA-binding protein PCBP1, allowing stabilisation of RNAs encoding the cytokines granulocyte-macrophage colony-stimulating factor (GM-CSF) and IL-2.

However, the most compelling evidence demonstrating the critical role of iron in adaptive immunity comes from the recent characterization of a rare inherited combined immunodeficiency. Jabara, et al. [186] described two families from Kuwait and Saudi Arabia in whom several affected children presented with recurrent, in some cases fatal, infections. Haematological parameters including lymphocyte counts were essentially normal, although several of the children displayed hypo- or a-gammaglobulinaemia. The affected individuals shared a hypomorphic mutation in the gene encoding TfR1 (*Tfrc*^Y20H/Y20H^) that caused impaired cellular iron acquisition as TfR1 internalization was abrogated. This defect markedly inhibited T/B cell proliferation and antibody class-switching, accounting for the immunodeficiency. Provision of non-transferrin-bound iron rescued lymphocyte proliferation ex vivo. These data showed that adaptive immunity is highly compromised when lymphocytes are unable to access iron [186].

This inherited combined immunodeficiency is rare; however, the low serum iron concentrations observed during the first year of life in Gambian infants [17]—a time when several Expanded Programme of Immunisation (EPI) vaccines are administered—may occur in millions of infants across LMIC settings. Lack of available serum iron could analogously limit the ability of lymphocytes to acquire iron to support clonal expansion and differentiation of adaptive responses. Whether this contributes to impaired vaccine responses [161,162], and whether adaptive immune responses can be enhanced through iron interventions around the time of vaccination should be investigated. More generally, how iron status and iron interventions influence immune ontogeny during early life should also be assessed, preferably in a variety of settings (urban, rural, malaria-endemic), ideally in combination with long-term outcomes.

### 3.5. Iron and Infection

While evidence that iron restriction may limit adaptive immune responses increases, strong epidemiological, experimental and evolutionary evidence that iron limitation is a protective strategy against some infections has accumulated (reviewed extensively elsewhere, e.g., [187,188,189]).

#### 3.5.1. Iron Status and Malaria Risk

Nutritional ID in young children (the identification of which typically relies on the interpretation of the acute phase protein serum ferritin) was found to be associated with reduced risk of subsequent malaria in multiple studies from distinct malaria-endemic regions within Africa: namely Tanzania (children aged 0–3 years; reduced parasitaemia, severe disease and mortality) [190], Malawi (6–60 months; reduced parasitaemia and clinical malaria) [191], Uganda (1–5 years) and Kenya (0–7 years; reduced risk of malaria episodes) [125], besides an earlier study in Kenya (8 months–8 years; reduced clinical malaria risk) [192]. Ex vivo *Plasmodium falciparum* cultures show that erythrocytes from iron-replete individuals are favoured for invasion compared to those from iron deficient individuals, and that supplementation of individuals with ID makes their erythrocytes more permissive to invasion in vitro [193]. This effect likely reflects the preferential invasion of reticulocytes and young erythrocytes by *Plasmodium falciparum* merozoites: by definition, iron interventions that effectively correct anaemia create such preferred targets for parasite invasion. Thus, it is plausible that a mild iron deficiency can represent an adaptive state protecting against malaria and/or other infections.

Iron intervention trials in similar settings have found increased risk of malaria and mortality in children administered therapeutic iron. The most prominent amongst these was a large trial in Pemba, Tanzania, that was halted early due to increased deaths and hospital admissions, both malaria- and non-malaria-related, among children given iron-containing supplements [194]. No such effect was found in another supplementation trial in Nepal (an area without endemic malaria) [195], yet other studies have also reported increased malaria risk in iron-supplemented iron deficient children [196,197]. The importance of the co-administration of malaria prophylaxis with iron interventions is revealed by stratified analysis of a Cochrane review: although no overall effect of iron interventions on malaria risk was found, malaria risk was increased if no control measures were available [198]. These findings underlie the WHO recommendations that iron interventions should only be administered when concurrent measures for diagnosis, prevention and treatment are in place (discussed in detail in [124]).

However, interactions between iron and infective agents are not restricted to malaria. Iron administration was also associated with increased risk of respiratory infections, and of diarrhoea in a large randomised controlled trial (RCT) in Pakistan [199]. Oral iron supplementation influences intestinal microbiome composition, favouring the outgrowth of more pathogenic bacteria over commensals, and driving inflammatory gut phenotypes.

#### 3.5.2. Iron and the Microbiome

Similarly to brain and immune development, critical changes in microbiome development occur during the first few years of life [169]. Indeed, it is becoming clear that these systems are not mutually exclusive. For example, changes in microbiome composition can influence systemic immunity [170]. Variability in microbiome development in infancy is driven by factors including mode of delivery, breast- versus formula-feeding, nature of weaning diets, exposure to antibiotics and geographic setting [169].

Early studies assessing the efficacy of iron interventions tended not to include systematic assessment of morbidity. However, an early systematic review and meta-analysis evaluating the effect of iron supplementation and fortification on infectious diseases did identify increased diarrhoea risk with interventions [200], corresponding to an extra 0.05 episodes/child year. Several more recent studies have likewise reported increased diarrhoea risk with iron supplementation and provision of multiple micronutrient powder (MNP, containing iron equivalent to 12–15 mg/day) in young children, notably in Bangladesh [201], Peru [202] and Ghana [203] (reviewed in [204]). The large study in Pakistan described above similarly reported increases in diarrhoea and bloody diarrhoea prevalence in the groups receiving iron-containing MNPs compared to control [199]. This study was blinded within the two MNP groups (with and without zinc), but the control group was unblinded to the intervention meaning the authors could not fully exclude reporter bias in collection of self-reported morbidity data. Nonetheless, an increase in bloody diarrhoea incidence of ~0.08 episodes per child year was reported, equivalent to approximately one additional bloody diarrhoea episode per year for every 12–13 children administered MNPs containing iron [199,204]. Furthermore, a meta-analysis of studies also identified that iron supplementation was associated with vomiting and fever [144].

Modification of the colonic microbiota ecosystem potentially underlies the iron-associated increase in diarrhoea prevalence in LMIC settings [205]. Iron is a required nutrient for almost all microorganisms, with a few notable exceptions (e.g., *Lactobacillaceae*, which utilize manganese) [206]. Bacteria that are well-adapted for acquiring iron may experience a fitness advantage during iron supplementation, as shown elegantly for the probiotic strain *E. Coli* Nissle, which effectively outcompetes *Salmonella* for iron, displacing it from its ecologic niche [207]. Many conditional pathogenic bacterial species have evolved elaborate iron acquisition mechanisms (e.g., siderophore production), enabling thriving at low iron concentrations (e.g., 10–30 µM in blood). These mechanisms may similarly confer growth advantages in environments richer in iron: faecal iron concentrations in adults and weaned infants may reach 1800 µM [205]. In contrast, human milk typically contains <10 µM iron [7,204], meaning iron concentrations in the gut in pre-weaning infants are likely considerably lower. At weaning, iron concentrations are likely to increase, potentially affecting microbiota composition, and are associated with barrier function and immune stimulating properties.

Within the intestinal microbiome, *Bifidobacteria* spp. and *Lactobacilli* spp. are generally associated with increased short chain fatty acid production and improved gut barrier function (highly desirable in infancy, particularly in resource-poor settings), while *Enterobacteriaceae* may include several potentially pathogenic species. Several recent studies in African LMICs have found increases in *Enterobacteriaceae* spp. with concurrent decreases in *Lactobacilli* spp. and *Bifidobacteria* spp. following provision of iron interventions [157,158,208,209]. Zimmermann, et al. reported such effects together with increased faecal calprotectin (marking gut inflammation) in Ivorian children receiving 20 mg Fe/day [209]. In infants administered MNPs containing 12 mg Fe/day, Jäggi, et al. similarly observed raised faecal calprotectin coupled with increased pathogenic *E. coli* abundance [157]. The effect on calprotectin was not detected at a lower dose (2.5 mg Fe/day), or in a subsequent trial using 5 mg Fe/day [158]. However, the latter trial confirmed reduced *Bifidobacterium* spp. and *Lactobacillus* spp. abundance, and also reported increased expression of virulence and toxin genes and fatty acid binding protein, a marker of enterocyte damage, following iron administration [158].

The effects of iron on the microbiome may be strongly setting specific. In the Kenyan study described above [157], baseline prevalence of potential enteropathogens (e.g., *Bacillus cereus*, *Streptococcus aureus*, *Clostridium difficile*) was very high. In contrast, a study of iron interventions to South African school children living an area with improved water supply, and generally better hygiene and sanitation, did not observe effects on gut microbiota with high-dose therapeutic iron (50 mg Fe/day) [210]. Similarly, the administration of iron drops to Swedish infants was associated with decreased *Bifidobacteria* spp. and increased *Clostridium* spp. abundance, but no enhanced growth of specific pathogenic bacterial strains, or any effect on faecal calprotectin; at the same time, provision of high iron formula decreased *Lactobacilli* abundance [211].

More nuanced approaches for iron supplementation in resource-poor settings may be effective and practicable. For example, provision of prebiotic fibres such as galacto-oligosaccharides (GOS, reminiscent of human milk oligosaccharide (HMO) structures) may counteract iron-associated perturbations of the gut microbiota [158]. This approach was also coupled with measures to improve iron bioavailability such as the use of ascorbic acid, NaFeEDTA (ferric sodium ethylenediaminetetraacetate) and an exogenous phytase enabling substantial reduction of iron dosing while maintaining efficacy [158]. The remarkable effects of GOS on iron bioavailability [212] require further investigation; specifically, how different structures and sources of fibers with similarities to GOS influence gut microbiota and iron absorption in infants should be studied.

#### 3.5.3. Iron and the Hepcidin Response to Infection/Inflammation

While baseline iron status and iron interventions may influence malaria risk and microbiome profiles, perturbations in host iron homeostasis occur as part of the response to infection, with a central role for hepcidin likely [114,213]. Hepcidin upregulation is observed during many important human infections including uncomplicated malaria [67,214,215], HIV-1 [216], *Mycobacterium tuberculosis* [217] and HIV-associated tuberculosis [218], and *Salmonella* Typhi [219] infections with accompanying hypoferremia or anaemia. It is likely that childhood infections similarly encompass hepcidin induction. Indeed, associations between elevated hepcidin and respiratory infection or fever, but not diarrhoea, in young children were recently reported [90].

Various consequences of hepcidin induction during infection have been shown using mouse models. Hepcidin-mediated iron redistribution during blood-stage parasitaemia in murine malaria models limited susceptibility to secondary liver stage infections; modelling suggested that this effect could explain why young children in areas of high malaria transmission tend to carry higher parasite densities but with fewer parasite genotypes than older children [220]. Hepcidin-mediated limitation of serum iron protects against extracellular siderophilic bacterial infections [69,70], most likely through limiting availability of non-transferrin bound iron (NTBI) to the bacteria. In contrast, macrophage iron retention due to hepcidin activity may favour the proliferation of intracellular macrophage-tropic infections such as *Salmonella* Typhimurium, although not through limiting bacterial iron acquisition, but by reduced generation of vacuolar antimicrobial ROS [221,222].

Another context where hepcidin upregulation may have a key protective role in the context of infancy relates to physiological neonatal hypoferremia. Despite neonates typically being iron replete and cord blood containing a high serum iron concentration, a significant hypoferremia is described in the hours following birth, lasting for a couple of days, consistent with hepcidin activity [82,83]. Neonatal septicaemia is a major contributor to infant mortality, especially in resource-limited settings. A recent study demonstrated significantly enhanced the growth of exemplar sentinel bacterial in sera taken from adults 4 h post-iron-supplementation (2 mg/kg), compared to pre-supplementation, and importantly that growth rate was strongly positively associated with transferrin saturation, irrespective of supplementation status [223]. Thus, variations in the degree of hypoferremia induced immediately after birth could influence the likelihood of septicaemic infections becoming established.

#### 3.5.4. Infection and Inflammation, and Iron Utilisation

Recurrent or chronic infections, or other drivers of persistent inflammation, even if mild, may also lead to raised hepcidin and relative inhibition of dietary iron absorption [90,106]. The specific relationship between hepcidin upregulation and iron absorption may be context dependent. A recent study found that an acute mild inflammatory response to vaccination is associated with raised hepcidin and decreased serum iron, but no detectable difference in iron absorption in Moroccan women with and without IDA [224]. It is conceivable that dietary iron absorption and macrophage iron release are differentially sensitive to mildly raised hepcidin, or that during hypoferremia, enterocyte cellular iron metabolism (IRP/IRE and/or HIF-2alpha) compensates for the inhibitory effect of hepcidin on mucosal iron transport. Nevertheless, chronic (even mild) infection and inflammation are still likely to contribute to the burden of iron deficiency through persistent hepcidin-mediated impairment of iron absorption. In line with this principle, treatment of asymptomatic *Plasmodium* parasitaemia in Beninese women and Ivorian school-age children reduced mild inflammation, normalised hepcidin concentrations, with consequent increases in iron absorption [225,226]. Targeting infections, including malaria, and limiting inflammation are therefore likely very important not only as ends in themselves, but also as interventions for iron deficiency and adjuncts to effective oral iron-based interventions [124,227].

## 4. Interventional Strategies: How and Should Iron Status Be “Improved”?

### 4.1. Risk-Benefit Assessments

To summarise the discussion above, iron is critical for erythrocyte function and oxygen carriage, for brain development, in supporting immune function and, more generally, for numerous cellular processes. Although treating or preventing iron deficiency gives clear haematological benefits, it may concurrently increase risk of infections such as malaria, or those involving gastrointestinal and respiratory systems. Furthermore, delivery of iron to iron-replete children may be associated with reduced growth, and a risk of toxicity when iron is in excess. The issue of whether to administer iron at preventive or therapeutic doses in LMIC settings with limited sanitary and health infrastructure therefore presents a classic risk-benefit problem. Systematic risk-benefit assessments including data-based simulations have been initiated [228], while large, rigorous randomised-controlled trials such as BRISC (Benefits and Risks of Iron interventionS in Children) promise to provide further insight towards guidelines for iron administration in young children [149]. Nonetheless, it may be unlikely that a single recommendation will ever find global validity; approaches that are adapted to the geographical settings the population and ideally at the individual level, are likely needed.

Iron supplementation, industrial food fortification with iron, and point-of-use fortification with iron-containing multiple micronutrient powders are some of the approaches recommended to address the anaemia burden. These interventions generally increase ferritin and haemoglobin concentrations and reduce anaemia prevalence [229,230,231,232,233]. However, despite examples of efficacy of fortification programs and MNP/iron-supplement interventions [234,235], anaemia persists as a global health problem. The reasons are clearly multifactorial, including lack of political awareness and recognition, programmatic limitations, uncertainty of the accuracy of population iron status assessment [236], besides lack of efficacy in the presence of infection or inflammation, and because a large proportion of anaemia is not iron-responsive in high-risk populations [1,124]. Moreover, caution should be employed in administering high therapeutic iron doses in settings with high infection burden. In line with principles for tackling anaemia of chronic disease [237], underlying infections should be treated first. Indeed, this approach alone may increase erythropoiesis and haemoglobin levels even when iron itself is not administered [124,225,238]. Correspondingly, raised haemoglobin and iron status was found to correlate with a documented interruption in malarial transmission in highland Kenya [239].

### 4.2. Optimisation of Iron Intervention Strategies

Global recommendations for providing iron supplements have evolved from a view of indiscriminately providing iron at high dosage (2 mg Fe/Kg body weight) to all infants and children if anaemia prevalence at 1 year of age in a population exceeds a given threshold (e.g., 40%) [240]. Recent recommendations stress the multifactorial nature of nutritional anaemia [241], recognising that control programs need to include multiple components. Currently suggested supplementation regimens for children aged 6–23 months are lower in iron dosages (10–12 mg/day, ~1.1–1.4 mg/Kg body weight [241]) than previously recommended. However, besides improving iron status, these doses have still been associated with detrimental effects [194,199]. Approaches which integrate knowledge of local risk factors with known biological, nutritional and epidemiological contributors to anaemia are needed, including addressing the underlying causes of ID and anaemia.

Given that detrimental effects of iron may be dose dependent, lower or intermittent dose schemes which have higher fractional bioavailability [105,242] may yield fewer side effects [243]. Lower dosing has been shown to be effective in elderly patients and for maintaining iron status in regular blood donors [244,245]. These approaches have been tested experimentally in high resource settings and may similarly improve the safety and tolerability of therapeutic iron doses in lower-resourced settings.

Preventing ID onset through dietary approaches is highly desirable, and probably safer than universal therapeutic supplementation because of the lower iron doses involved. Such approaches should be promoted, recognising that diets with high iron bioavailability may not always be economically feasible in many LMICs. Fortification approaches with low yet bioavailable iron levels may be a prudent and realistic approach, and some food-based examples pointing in this direction have been recently published [246,247]. Alternative novel approaches to enhance iron bioavailability from foods, such as the use of GOS described above, show promise, particularly when they also promote the growth of barrier function bacteria [158,212]. Increasing consumption of ascorbic acid-rich foods likewise improves iron status [248], suggesting that dietary approaches to improve iron status do not necessarily need to include increased dietary iron content.

### 4.3. Maternal Iron-Based Interventions

Furthermore, prevention of childhood ID provides an additional rationale for investing in measures aimed at preventing preterm births and increasing birthweight—including promotion of delayed cord clamping and maternal iron supplementation to prevent IDA during pregnancy. As discussed, longer gestation, higher birthweight and delayed cord clamping lead to reduced ID risk later in infancy [7]. An RCT of iron supplementation (60 mg daily) during pregnancy in a malaria-endemic setting in Kenya found no increased risk of malaria infection when iron supplementation was given, but increased birthweight and length of gestation; effects were more notable in women with baseline ID [249]. Similar trends have also been reported across other trials of iron interventions during pregnancy [250]. Thus, ensuring positive birth outcomes provides a further strategy for enhancing childhood iron status, apparently without the same infection risks that are associated with early childhood therapeutic iron administration.

## 5. Concluding Remarks

In conclusion, defining iron status in settings with high infection and inflammation remains a challenge. Similarly, the biological understanding of optimal iron status for specified ideal functional outcomes (neurological, immunological, physical) remains limited despite the rapid and substantial advances in the understanding of iron biology experienced in the last two decades. Translating markers of positive functional outcomes to clinically and field-suitable definitions of iron status, balanced against known side-effects and risks of “improving” iron status in young children in such settings, presents a challenge to be tackled by the iron and health community in the coming years (Box 2). Programmatically, it will be desirable for iron interventions to be adapted to the local sanitary and dietary circumstances and to contain iron levels appropriately targeted to the prevailing iron status. While dietary and fortification approaches appear safer due to the lower iron levels involved, higher-dose therapeutic and supplemental approaches may be appropriate in well-defined settings of highly prevalent deficiency with developed/controlled sanitary conditions, and when they can be individually targeted. In areas affected by high burdens of infection/inflammation and iron deficiency, strategies aimed at improving iron status and anaemia prevalence without the necessity of administering interventional iron to children should be thoroughly evaluated. These include direct targeting of the causes of infection/inflammation, the use of enhancers of iron absorption, and through maternal iron interventions during pregnancy which appear safer and should benefit both maternal and infant outcomes.

Box 2Research questions and priorities.Defining iron status in young childrenPriority: standardisation of assays for hepcidin and sTfR.Can point-of-care diagnostics for ferritin, sTfR and other analytes including hepcidin be developed, optimised and made cost-effective for use in LMIC settings?What is the optimal non-invasive definition of iron deficiency in infancy?Can the mechanistic understanding of iron homeostasis in infancy and early childhood be harnessed to define cut-offs of deficiency (e.g., haemoglobin, ferritin, hepcidin, erythrocyte markers such as MCV) in infancy and early childhood?Is hepcidin an optimal biomarker of iron absorption/utilisation in young children?Is it more important to classify iron deficiency, or the ability to safely and efficiently absorb iron?Iron in key physiological processes of early childhood
How is iron prioritised between the brain, bone marrow and other iron-demanding tissues during different degrees of iron depletion, and at different stages of infancy/early childhood?Is there a hierarchy of sensitivity of different cellular processes (e.g., mitochondrial function, nucleotide synthesis) to cellular iron deficiency?Are erythroferrone or other hepcidin-suppressive proteins involved in iron handling during the iron-demand of early childhood?What are the main drivers of low-grade inflammation in LMIC settings, and are these associated with raised hepcidin and impaired iron utilisation?Is there a beneficial effect of iron interventions on cognitive outcomes in infants in LMIC settings? Larger, high quality trials are likely required to establish this, and the potential of supplementing iron early in post-natal life should be considered.Does iron reduce child growth when given to iron-replete infants and, if so, by what mechanism?How does iron deficiency influence both innate and cell-mediated adaptive immune responses to infections and vaccines?Do iron-associated microbiome changes associate with changes in systemic phenotypes including immunity and brain development?Will higher resolution microbiome analysis of iron-associated changes yield useful information on specific iron-related effects?Interventions aimed at adjusting iron status
Can iron status be improved in LMIC settings in the absence of exogenous iron interventions? E.g., by treating infection/inflammation; by increasing bioavailability using absorption enhancers?Is there an optimal combination of dietary components (e.g., phytase, ascorbic acid, organic acids, muscle protein, GOS, other dietary fibres) that can enhance bioavailability while reducing iron dosage to facilitate supplementation without microbiome-associated adverse effects?Could strategies for delivering haem-iron therapeutically be broadly implemented?Is there a role for intravenous iron in addressing disturbed iron status in LMIC settings?Are antenatal iron interventions and perinatal interventions (e.g., delayed cord clamping) effective in improving neonatal outcomes, including iron status later in infancy, across varied LMIC settings?Can setting-specific recommendations be made for the likely relative risks and benefits of giving iron, and the likely optimal mode of delivering iron?Is there benefit from programmatically promoting dietary counselling relating to iron intake and bioavailability enhancement?

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
