# Peer review of "The Importance of Iron Status for Young Children in Low- and Middle-Income Countries: A Narrative Review"

_pharmaceuticals, 2019, doi:10.3390/ph12020059_

Reviewer 1 Report

The review of Armitage and Moretti deals with iron status for young children in low and middle-income countries. The importance of iron nutrition is recognized in the context of child nutrition and more detailed studies can also promote meaningful advances and favor new proposals to tackle the problem. The review has the merit of addressing the iron status for young children in a general contest not limited only to a region or nation or geographical area. The review structure is complete. The first part dealing with the biochemical aspect of iron is concise but recalls several essential points. Obviously, given the vastness of the topic, the review cannot address this subject in more detail, but sufficient information about iron trafficking is presented and results useful both to those who are familiar with iron biochemistry but above all to those who do not know the elementary aspects of this topic. The review is schematic also in the second part in the presentation of iron deficiency effects in cognitive processes and other pathologies, and in possible interventions to address the iron deficiency from a diagnostic point of view and then therapeutic.

The paper deserves publication after two minor points.

Minor concerns:

Add in the reference section the recent paper “Rapid diagnostics for point-of-care quantification of soluble transferrin receptor” by Srinivasan B et al, EBioMedicine 2019, about soluble transferrin receptor.

The authors correctly mention the references of Larsen et al by the way The authors should make it clear in the paper that few large, placebo-controlled trials in low-income countries have been performed to assess the iron supplementation on cognition. It is necessary to have more data to justify policies of universal iron intervention in young children.

Author Response

Comments and Suggestions for Authors

The review of Armitage and Moretti deals with iron status for young children in low and middle-income countries. The importance of iron nutrition is recognized in the context of child nutrition and more detailed studies can also promote meaningful advances and favor new proposals to tackle the problem. The review has the merit of addressing the iron status for young children in a general contest not limited only to a region or nation or geographical area. The review structure is complete. The first part dealing with the biochemical aspect of iron is concise but recalls several essential points. Obviously, given the vastness of the topic, the review cannot address this subject in more detail, but sufficient information about iron trafficking is presented and results useful both to those who are familiar with iron biochemistry but above all to those who do not know the elementary aspects of this topic. The review is schematic also in the second part in the presentation of iron deficiency effects in cognitive processes and other pathologies, and in possible interventions to address the iron deficiency from a diagnostic point of view and then therapeutic.

The paper deserves publication after two minor points.

Response:We thank the reviewer for this assessment. We have responded to the two points raised.

Minor concerns:

Add in the reference section the recent paper “Rapid diagnostics for point-of-care quantification of soluble transferrin receptor” by Srinivasan B et al, EBioMedicine 2019, about soluble transferrin receptor.

Response:We have now added the following to the section “Classification of iron deficiency / Markers of iron-restricted erythropoiesis”:

“Of note, paper-based smartphone-app-guided point-of-care assays for ferritin and sTfR show promise in proof-of-principle studies, suggesting field measurement of these markers in children from resource-limited settings may become feasible[97,98]

We also added the reference corresponding to the POC test for ferritin, previously reported by the same group.

Finally, we added the following to “Box 2: Research questions and priorities”:

·       “Can point-of-care diagnostics for ferritin, sTfR and other analytes including hepcidin be developed, optimised and made cost-effective for use in LMIC settings?”

The authors correctly mention the references of Larsen et al by the way The authors should make it clear in the paper that few large, placebo-controlled trials in low-income countries have been performed to assess the iron supplementation on cognition. It is necessary to have more data to justify policies of universal iron intervention in young children.

Response:We have now modified the section “Iron and neurological development / Iron interventions and cognitive outcomes”:

….In contrast, several studies indicate benefits of iron supplementation on cognitive function in older children [145-147], in particular following longer-term supplementation. [MOVED UP] Within this discussion, it should also be borne in mind that iron replete Chilean infants who received high iron formulas displayed reduced cognitive performance at 10 years of age[148].

Together, these analyses highlight the general physiological importance of correcting ID for cognitive development, while at the same time calling for further detailed investigation during early perinatal life. Indeed, few high quality adequately-powered placebo-controlled trials in low-income countries aimed at assessing the impact of iron interventions on cognitive function have been performed [146][LARSON ref]. Such data would be highly desirable for informing global health policies regarding universal iron intervention in young children; one such trial - BRISC (“Benefits and Risks of Iron interventionS in Children”) - is currently ongoing in Bangladesh [149][HASAN ref].

Reviewer 2 Report

The authors have done a great job. The truth that is worthy of the chapter of a book. I think they're going to give the journal a lot of visibility. From the methodological point it is fantastically organized. However, I would consider some details.

Since it is a narrative review, I believe that the authors should indicate it in the title.

The abstract should be restructured indicating what is its final conclusion of the review, since it is an aspect that the authors do not indicate. The same should be specified at the end of the text. That is, what conclusion is reached in relation to the objectives set.

On the other hand, although I understand that it should be a matter of editing, the boxes should be in a box format, not just in cursive.

Author Response

Comments and Suggestions for Authors

The authors have done a great job. The truth that is worthy of the chapter of a book. I think they're going to give the journal a lot of visibility. From the methodological point it is fantastically organized.

Response:We thank the reviewer for the kind comments – they are appreciated.

However, I would consider some details.

Since it is a narrative review, I believe that the authors should indicate it in the title.

Response:We respectfully point out that the other reviews written for this special issue on iron do not seem to state explicitly in the titles that they are narrative reviews. We therefore request to retain the current title if possible please. However, we have clarified explicitly in the abstract and introduction that this article is a narrative review.

Abstract: “…In this narrative review, we first examine demand and supply of iron during early childhood, in relation to molecular understanding of systemic iron control. We then evaluate the importance of iron for distinct aspects of physiology and development, particularly focusing on young LMIC children. We finally discuss the implications and potential for interventions aimed at improving iron status whilst minimising infection-related risks in such settings.”

Introduction: “In this narrative review, we discuss the biology of iron in early childhood, with a particular focus on children living in settings with high burdens of infection, nutritional deficiency and anaemia.”

The abstract should be restructured indicating what is its final conclusion of the review, since it is an aspect that the authors do not indicate. The same should be specified at the end of the text. That is, what conclusion is reached in relation to the objectives set.

Response:We have modified the abstract accordingly as follows:

Early childhood is characterised by high physiological iron demand to support processes including blood volume expansion, brain development and tissue growth. Iron is also required for other essential functions including generation of effective immune responses. Adequate iron status is therefore a prerequisite for optimal child development, yet nutritional iron deficiency and inflammation-related iron restriction are widespread amongst young children in low- and middle-income countries (LMICs), meaning iron demands are frequently not met. Consequently, therapeutic iron interventions are commonly recommended. However, iron also influences infection pathogenesis: iron deficiency reduces the risk of malaria, while therapeutic iron may increase susceptibility to malaria, respiratory and gastrointestinal infections, besides reshaping the intestinal microbiome. This meanscaution should be employed in administering iron interventions to young children in LMIC settings with high infection burdensIn thisnarrative review, we first examine demand and supply of iron during early childhood, in relation to molecular understanding of systemic iron control. We then evaluate the importance of iron for distinct aspects of physiology and development, particularly focusing on young LMIC children. We finally discuss the implications and potential for interventions aimed at improving iron status whilst minimising infection-related risks in such settings. Optimal iron intervention strategies will likely need to be individually or setting-specifically adapted according to iron deficiency, inflammation status and infection risk, while maximising iron bioavailability and considering the trade-offs between benefits and risks for different aspects of physiology. The effectiveness of alternative approaches not centred around nutritional iron interventions for children should also be thoroughly evaluated: these include direct targeting of common causes of infection/inflammation, and maternal iron administration during pregnancy.

We have modified the concluding paragraph as follows:

Concluding remarks

In conclusion,defining iron status in settings with high infection and inflammation remains a challenge. Similarly,the biological understanding of optimal iron status for specified ideal functional outcomes (neurological, immunological, physical) remains limited, despite the rapid and substantial advances in the understanding of iron biology experienced in the last two decades. Translating markers of positive functional outcomes to clinically and field-suitable definitions of iron status, balanced against known side-effects and risks of “improving” iron status in young children in such settings, presents a challenge to be tackled by the iron and health community in the coming years (Box 2). Programmatically, it will be desirable for iron interventions to be adapted to the local sanitary and dietary circumstances and to contain iron levels appropriately targeted to the prevailing iron status. While dietary and fortification approaches appear safer due to the lower iron levels involved, higher-dose therapeutic and supplemental approaches may be appropriate in well-defined settings of highly prevalent deficiency with developed/controlled sanitary conditions, and when they can be individually targeted. In areas affected by high burdens of infection/inflammation and iron deficiency, strategies aimed at improving iron status and anaemia prevalence without the necessity of administering interventional iron to children should be thoroughly evaluated. These include direct targeting of the causes of infection/inflammation, the use of enhancers of iron absorption, and through maternal iron interventions during pregnancy which appears safer and should benefit both maternal and infant outcomes.

On the other hand, although I understand that it should be a matter of editing, the boxes should be in a box format, not just in cursive.

Response:We have now placed the text within boxes as requested, rather than just writing as free text.

Round  2

Reviewer 2 Report

The authors have made a great effort to answer all the comments proposed by the reviewer. However, I insist that although the text is well described as a bibliographic revision, this term should be in the title. Therefore, I consider that the title should be: "The importance of iron status for young children in low- and middle-income countries: A narrative review”.

Author Response

Thank you once again to the reviewer for taking time to consider our review article.  

We have made the change to the title as requested to: "The importance of iron status for young children in low- and middle-income countries: a narrative review."